# Low-Field Benchtop NMR to Discover Early-Onset Sepsis: A Proof of Concept

**DOI:** 10.3390/metabo13091029

**Published:** 2023-09-21

**Authors:** Matteo Stocchero, Claire Cannet, Claudia Napoli, Elena Demetrio, Eugenio Baraldi, Giuseppe Giordano

**Affiliations:** 1Women’s and Children’s Health Department, University of Padova, 35128 Padova, Italy; eugenio.baraldi@unipd.it (E.B.); giuseppe.giordano@unipd.it (G.G.); 2Fondazione Istituto di Ricerca Pediatrica Città della Speranza, 35127 Padova, Italy; 3Bruker BioSpin GmbH, 76275 Ettlingen, Germany; 4Bruker Italia S.r.l., 20158 Milano, Italy; claudia.napoli@bruker.com (C.N.);

**Keywords:** low-field benchtop NMR, fingerprinting, metabolomics, sepsis

## Abstract

Low-field (LF) benchtop NMR is a new family of instruments available on the market, promising for fast metabolic fingerprinting and targeted quantification of specific metabolites despite a lack of sensitivity and resolution with respect to high-field (HF) instruments. In the present study, we evaluated the possibility to use the urinary metabolic fingerprint generated using a benchtop LF NMR instrument for an early detection of sepsis in preterm newborns, considering a cohort of neonates previously investigated by untargeted metabolomics based on Mass Spectrometry (MS). The classifier obtained behaved similarly to that based on MS, even if different classes of metabolites were taken into account. Indeed, investigating the regions of interest mainly related to the development of sepsis by a HF NMR instrument, we discovered a set of relevant metabolites associated to sepsis. The set included metabolites that were not detected by MS, but that were reported as relevant in other published studies. Moreover, a strong correlation between LF and HF NMR spectra was observed. The high reproducibility of the NMR spectra, the interpretability of the fingerprint in terms of metabolites and the ease of use make LF benchtop NMR instruments promising in discovering early-onset sepsis.

## 1. Introduction

NMR spectroscopy has been successfully applied in the last 40 years to investigate the metabolic content of biofluid mixtures, such as urines and plasma, in order to discover biomarkers and reveal hidden biochemical mechanisms underlying complex diseases [1,2]. Due to its high level of reproducibility and stability, simple and non-destructive sample preparation, short time of analysis and the possibility to simultaneously identify and quantify a large number of metabolites, high-resolution high-field (HF) NMR spectroscopy has become a standard approach in targeted and in untargeted metabolomics. Unfortunately, the high cost of the instrumentation, its large size, the use of cryogenic systems and the need of a specialist technical staff limited its accessibility to many researchers. In the last 10 years, the progress in the technology underlying magnets has led to permanent magnets with a magnetic field sufficiently homogeneous to allow NMR spectroscopy using low-field (LF) [3,4]. A new family of instruments for NMR spectroscopy operating at less than 2 T was available on the market. Interestingly, the new instruments are benchtop, do not require complex operations for setting and, then, dedicated expert users; they are easy to maintain, and it is also easy to achieve low running cost. In spite of a lack of sensitivity and resolution with respect to HF instruments, benchtop LF NMR instruments seem to be suitable for metabolic fingerprinting and targeted quantification of specific metabolites, fitting the needs of clinical applications [5,6].

Another standard approach to metabolomics is based on Mass Spectrometry (MS) [7,8]. A large number of techniques have been developed combining chromatography and Mass Spectrometry to identify and quantify metabolites in complex biological samples. These techniques outperform HF NMR spectroscopy in terms of sensitivity using moderately expensive instrumentation with small footprint, but require complex sample preparation, show moderate reproducibility and lead to partial structure determination.

The aim of our proof-of-concept study was to evaluate whether a benchtop LF NMR instrument may be promising for an early detection of sepsis by urinary metabolic fingerprinting at birth, and to compare the classifier based on its fingerprint with that obtained with the more complex MS-based platform.

Neonatal sepsis is an infection-induced systemic inflammatory response syndrome common in premature and term neonates [9,10,11]. It is one of the leading causes of neonatal death and morbidity and is believed to have a key role in most inflammatory disorders that cause or enhance the main morbidities affecting the preterm (bronchopulmonary dysplasia, white matter injury, necrotizing enterocolitis, and retinopathy of prematurity). Sepsis in newborns is typically classified as either early-onset sepsis (EOS), when the infection occurs within three days after birth, or late-onset sepsis (LOS) if it develops afterward. Early detection of neonatal sepsis and prompt administration of broad-spectrum antibiotic therapy can prevent its clinical course towards septic shock and death, but it is not easy to diagnose neonatal sepsis early on. Blood culture is still considered the gold standard, even though it takes time to obtain the results, and false-negative findings are not uncommon because neonatal bacteremia is often intermittent, and intrapartum antibiotic treatment may limit the culture’s diagnostic value [12]. Neonatal sepsis is therefore mainly suspected on the grounds of non-specific clinical signs and symptoms; moreover, none of the most widely used biomarkers are entirely reliable indicators of sepsis in newborns [13,14].

In Mardegan et al. [15], the possibility to apply untargeted MS-based metabolomics for an early detection of sepsis has been investigated. The study highlighted that neonates with EOS have a perturbation at the urinary metabolic level at birth that clearly distinguishes them from those without sepsis. Specifically, some metabolites belonging to glutathione and tryptophan pathways resulted to be promising as new biomarkers of neonatal sepsis. However, mass profiling requires highly trained personal and complex analytical instrumentation that make difficult direct translation of the method in a clinical environment.

Here, the same urine samples of that study were analyzed using a benchtop LF NMR instrument. Specifically, a fingerprinting approach was applied to prove the feasibility of building a tool able to detect EOS with a potential application in a clinical environment. We did not investigate the complex biochemical mechanisms underlying EOS as performed by MS because the detail in the structure of 1D NMR spectra from LF instruments is not adequate for a comprehensive untargeted metabolomics analysis and allows only the identification of a small set of metabolites with high concentration. Indeed, a large number of resonances overlap and crowding problems are encountered. However, we took advantage of the high reproducibility of the spectra and of their relatively high sensitivity to a large number of metabolites to build representative fingerprints of the samples. Moreover, the reduced need for maintenance and expertise in sample preparation, as well as in the use of the instrument, and the short time and cost of analysis make LF NMR-based classification models easy to translate in a clinical environment.

## 2. Materials and Methods

### 2.1. Study Population

The present study is based on the samples collected in the study of Mardegan et al. [15], where preterm neonates (<37 weeks of gestation) admitted to the Neonatal Intensive Care Unit at the Women’s and Children’s Health Department of Padova Hospital (Italy) were recruited from December 2015 to November 2017 in a case–control study designed to investigate the metabolomics differences between neonates developing EOS and controls. Specifically, the EOS group included any neonates classified as septic infants on the basis of clinical signs (cardiovascular or respiratory instability, neurologic signs) and laboratory findings obtained within 72 h from birth (leukopenia, leukocytosis, increased C-reactive protein, increased serum lactate), in accordance with the criteria established in 2010 at an expert meeting of the European Medicines Agency on neonatal and pediatric sepsis. To avoid any influence of gestational age and weight, each neonate diagnosed with EOS was matched with the next eligible newborn of similar gestational age and weight who did not manifest any infection within seven days of birth (control group). More details about the design of the study can be found in Mardegan et al. [15].

### 2.2. Sample Collection

At least 2 mL of urine was collected non-invasively within 24 h of birth by placing a cotton ball inside the newborn’s nappy and checking for the presence of urine every 30 min. After the neonate urinated, the cotton ball was placed in the barrel of a syringe and squeezed with the plunger to collect the absorbed urine in a container prewashed with methanol. Samples were stored at −80 °C. More details about sample collection can be found in Mardegan et al. [15].

### 2.3. Untargeted MS-Based Metabolomics Analysis

Untargeted metabolic profiling was performed in positive and negative ionization mode on an Acquity Ultra Performance Liquid Chromatography (UPLC) system (Waters MS Technologies Ltd., Manchester, UK) coupled to a Quadrupole Time-of-Flight (QToF) Synapt G2 HDMS mass spectrometer (Waters MS Technologies Ltd., Manchester, UK). Chromatography was performed using an Acquity HSS T3 (1.7 μm, 2.1 × 100 mm) column (Waters Corporation, Milford, CT, USA).

Quality control samples with different dilution factors (QCs) and a standard mix solution were used to assess reproducibility and accuracy during the analysis. Local calibration models obtained for the QCs with different dilutions were used to normalize the ion intensity detected. Probabilistic Quotient Normalization (PQN) was applied to remove the effects of the physiological urine dilution. Data were log-transformed and mean-centered prior to performing data analysis. More details about sample preparation, instrumental procedures and data pre-processing, can be found in Mardegan et al. [15].

### 2.4. NMR Analysis

Urine samples were prepared according to standard procedures as previously described [16]. Frozen urine samples were thawed at 4 °C and shaken before use. Then, 0.9 mL of urine was added to 0.1 mL of potassium phosphate buffer (pH 7.4) containing trimethylsilylpropionic acid-d4 sodium salt (TSP) and sodium azide. The mixture was homogenized, and 0.6 mL were transferred to a 5 mm NMR tube for analysis.

The benchtop LF NMR spectra were acquired on a Bruker Fourier 80 High-Definition (HD) system (Bruker BioSpin, Ettlingen, Germany) with a 1H spectrometer frequency of 80.112 MHz. The benchtop LF NMR measurements did not require the use of internal standards. Instead, an external standard (demineralized water) was used for quantitative analysis and spectra scaling by means of ERETIC (Electronic to Access In Vivo Concentration) and PULCON [17] principles. The experiments were based on noesypr1d (1H spectra with solvent suppression). To efficiently suppress the water signal, a preparation experiment consisting of a simple 1H spectrum with no suppression and 4 scans was performed for every sample to accurately define the frequency (O1p) to suppress in the spectra. Subsequently, the noesypr1d spectrum was measured with the following pertinent parameters: recovery delay (D1) of 4 s, acquisition time of 5 s, number of scans of 64, totalizing a measurement time of 10 min per sample. Moreover, a pre-saturation field of 15 Hz was used for the solvent signal suppression. The sample temperature during the experiments was 25.0 °C.

The acquired data were automatically processed. Specifically, Fourier Transform of the Free Induction Decay was performed applying an exponential window function with line broadening of 0.3 Hz; the phase of the resulting spectra had the zero-order phase adjusted whilst first-order correction was not performed; the baselines of the spectra were automatically corrected by a first-order polynomial function. The resulting spectra were referenced to 0 ppm according to the TSP peak position. Spectra were aligned using the CluPA algorithm [18]. The water region between 4.50 and 5.25 ppm was excluded. Intelligent bucketing with a minimum width of 0.04 ppm was applied. Data were normalized by PQN to remove the effects of the physiological urine dilution and mean-centered.

The Regions Of Interest (ROIs) useful to distinguish neonates developing sepsis from controls discovered by data analysis were characterised in terms of metabolites investigating the HF NMR spectra obtained for the collected samples by a Bruker Avance IVDr system (600 MHz) (Bruker BioSpin, Ettlingen, Germany). Samples were analyzed in full automation according to standard procedures as previously described [16] using a 600 MHz Bruker Avance III HD NMR spectrometer equipped with an automated sample changer SampleJet with a sample cooling and pre-heating station, a 5 mm inverse probe with z-gradient, automated tuning, a matching and cooling unit BCU-I. TopSpin 3.6 (Bruker BioSpin, Ettlingen, Germany) in combination with Bruker’s body fluid NMR methods package B.I. Methods 2.5 was used for fully automated acquisition and processing controlled by ICON NMR. The experiments performed were standardized 1D NOESY-presat experiments and 2D J-resolved experiments as previously described [16]. The NMR signals of the HF NMR spectra belonging to the ROIs were annotated by searching the Bruker database BBIOREFCODE.

### 2.5. Statistical Data Analysis

Demographic and perinatal characteristics as well as laboratory findings at birth were investigated by t-test and Mann–Whitney test in the case of normally and non-normally distributed data, respectively, and by chi-squared test for categorical data.

Principal Component Analysis (PCA) was applied for exploratory data analysis and outlier detection [19].

PLS for classification (PLS2C) [20] was used to build the classifiers useful to distinguish neonates developing EOS and controls. The number of score components of the PLS2C models was determined on the basis of the first maximum of the Matthew’s correlation coefficient calculated by 20 repeated 5-fold cross-validation under the condition to pass the permutation test (1000 random permutations) on the class response. Stability selection based on Variable Influence on Projection allowed the selection of the predictors most relevant for the classifier [21]. Specifically, binary matrix sampling was applied to the observations and to the predictors with probabilities equal to 0.7 and 0.5, respectively, to extract 500 sub-training sets that were used to build 500 PLS2C sub-models. The analysis of the selected predictors in the model population allowed us the selection of the most relevant predictors.

The data variation of the datasets generated by untargeted MS-based metabolomics was compared with that of the dataset obtained by LF NMR using orthogonal Wold’s two-block Mode A PLS (OPLS-W2A) and orthogonally constrained PCA (oCPCA) [22]. OPLS-W2A was applied to discover the joint data variation, while oCPCA was applied to highlight the unique data variation. The number of score components was assessed exploring the eigenvalue structure of the model.

Data analysis was performed by in-house R-functions implemented by the R 4.2.2 platform (R Foundation for Statistical Computing, Vienna, Austria).

## 3. Results

For one neonate of the sepsis group enrolled in the study of Mardegan et al. [15], the volume of the collected urine was not enough to allow NMR analysis, since the previous MS-based analysis required the whole sample. As a result, the control group was composed of 10 neonates and the sepsis group consisted of of 8 neonates. The demographic and perinatal characteristics and the laboratory findings at birth of the 18 neonates are reported in Table 1. Assuming a significance level of 0.05, no significant differences were discovered between the two groups.

Untargeted MS-based metabolomics led to two datasets, one composed of 2394 variables generated by negative ionization mode (NEG dataset) and another of 3224 variables arising from the positive ionization mode (POS dataset). Considering each dataset, no outliers were detected analyzing each group of neonates by PCA and assuming a significance level of 0.05 for the T2-test and the Q-test.

Applying PLS2C, the best model obtained for the NEG dataset showed two score components, Matthew’s correlation coefficient in fitting the data (MCC) equal to 0.892 (*p* = 0.012), Matthew’s correlation coefficient calculated by 20 repeated fivefold cross-validation (MCCcv) equal to 0.433 (*p* = 0.032), area under the Receiver Operating Characteristic curve in fitting the data (AUC) equal to 1.000 (*p* = 0.005) and area under the Receiver Operating Characteristic curve calculated by 20 repeated fivefold cross-validation (AUCcv) equal to 0.725 (*p* = 0.023). For the POS dataset, the best model showed two score components, MCC equal to 0.892 (*p* = 0.015), MCCcv equal to 0.316 (*p* = 0.041), AUC equal to 1.000 (*p* = 0.009) and AUCcv equal to 0.663 (*p* = 0.025). Both models wrongly classified one sample of the sepsis group in fitting, while two errors and three errors in cross-validation were obtained for the control and the sepsis groups, respectively. The Matthew’s correlation coefficients for the out-of-bag predictions calculated by stability selection were 0.333 and 0.217 for the NEG and the POS datasets, respectively.

By pre-processing the LF NMR spectra, a dataset (indicated as Fourier 80 dataset in the following) composed of 84 variables (ROIs) was obtained. In Figure 1, the 18 NMR spectra and the intervals used for bucketing the spectra into the 84 ROIs are reported. Assuming a significance level of 0.05, no outliers were detected applying the T2-test and the Q-test to the PCA model built on each of the two groups.

Prior to solving the classification problem, the data variation of the Fourier 80 dataset was investigated by OPLS-W2A and by oCPCA to evaluate the common data variation shared with the NEG and POS datasets and the unique data variation, respectively. Specifically, the Fourier 80 dataset was compared with the predictive part of the post-transformed PLS2C models; it includes the data variation useful to distinguish the two groups under investigation, obtained for the NEG and the POS datasets. Considering the NEG dataset, the OPLS-W2A model showed one parallel score component explaining the 9.8% of the total variance of the Fourier 80 dataset and two orthogonal score components, whereas the oCPCA model build on the sum of the residuals and the orthogonal part of the model presented two score components explaining a unique data variation equal to 77.6% of the total variance. The correlation between the parallel score component and the predictive component was 0.76. For the POS dataset, the model of the Fourier 80 dataset showed one parallel score component explaining the 7.3% of the total variance, whereas the analysis of the unique data variation discovered two score components explaining the 80.1% of the total variance. The correlation between the parallel score component and the predictive component was 0.70. In Figure 2, the parallel score components of the two OPLS-W2A models are reported using a color code according to the group of the sample. Interestingly, the components seem to be suitable to distinguish the two groups since most of the controls showed positive values, while most of the neonates developing sepsis negative values. As a consequence, we can expect that the Fourier 80 dataset may lead to classification models that performs similarly or, if the unique part of the Fourier 80 dataset is suitable to distinguish the two groups, better than those obtained by the NEG and the POS datasets.

Considering the Fourier 80 dataset, the best PLS2C model able to distinguish the two groups showed two score components, MCC equal to 0.775 (*p* = 0.031), MCCcv equal to 0.325 (*p* = 0.040), AUC equal to 0.950 (*p* = 0.018) and AUCcv equal to 0.700 (*p* = 0.028). One sample belonging to the control group and one belonging to the sepsis group were wrongly classified in fitting, while the same errors in cross-validation of the models obtained considering the NEG and the POS datasets were observed. The score scatter plot obtained post transforming the model is reported in Figure 3. As it can be observed, the points representing urine samples belonging to different groups lie in different regions of the plot.

Figure 4 reports the distributions of MCCcv and AUCcv for the classification models obtained considering NEG, POS and Fourier 80 datasets. As it can be seen, the three datasets led to models with similar performance in cross-validation and, as a consequence, we can expect that one does not outperform the other in predicting new observations.

Applying stability selection and assuming a significance level of 0.05, a subset of 16 relevant ROIs was obtained (Table 2). Interestingly, the signals in the relevant ROIs showed high signal-to-noise ratio, as it can be observed in Appendix A. The median of the Matthew’s correlation coefficient calculated considering the out-of-bag predictions was 0.433. Also, when considering the out-of-bag predictions during stability selection, the three datasets showed similar results.

LF NMR was used within a fingerprinting approach, and the structure of the spectra is not adequate to extract clear information about the metabolites responsible for the discrimination of neonates developing sepsis and controls. As a consequence, to allow an interpretation of the behavior of the classifier in decision making in terms of metabolites, the signals of the relevant ROIs in the LF NMR spectra were annotated through the analysis of the corresponding regions in the HF NMR data. This was possible because the two sets of spectra were highly correlated, as the heatmap of Figure 5 shows. The diagonal of the heatmap corresponds to signals recorded by both instruments that detected the same regions of signals, even if they used a different resolution and signal-to-noise ratio. Due to the high resolution of the HF NMR spectra, the characteristic signals of 2,3,4-trihydroxybutyric acid, 3,4-dihydroxybutanoic acid, d-glucose, d-serine, gluconate, hippuric acid, lactate, L-threonine, N-glycine, pseudo uridine, ribitol, kynurenic acid, myo-inositol, taurine and phenylalanine were found in the relevant ROIs.

## 4. Discussion

Mardegan et al. [15] proved that the complex infection-induced systemic inflammatory response syndrome of sepsis produces a perturbation of the urinary metabolome. Specifically, untargeted MS-based metabolomics was applied to discover the differences in the urinary metabolome between neonates developing sepsis and controls, highlighting that some metabolic pathways were perturbed. Thousands of metabolites were quantified to discover a small set of relevant ones. Since the reproducibility of the analytical performance is only moderate and usually limited to a small number of independent analytical sessions, untargeted data are in general unsuitable for clinical applications, and specific targeted methods must be developed to translate the findings into clinical tools. As a consequence, single targeted methods should be developed for the relevant metabolites discovered, if one would want to test the approach in a clinical environment.

LF NMR spectra map a smaller chemical space than that explored by untargeted MS-based metabolomics, and only metabolites with concentration greater than 10^−5^ M can be detected [23]. However, the high reproducibility of the NMR fingerprint and the robustness of the LF NMR instrumentation make LF NMR a candidate analytical technique for clinical applications, provided that the perturbations at the metabolomic level are captured by the fingerprint.

In principle, it is not guaranteed that a perturbation discovered by MS can be measured also by LF NMR, and this must be experimentally proven, as in our study. It is worth noting that the two approaches do not necessarily need to measure the same metabolites, since metabolic pathways are closely related to each other and a perturbation is usually not localized into a single pathway, but rather affects the concentration of a large number of metabolites, even far away from that pathway. Moreover, in the case of a multifactorial disease such as sepsis, several pathways are perturbed and a large perturbation is generated in the urinary metabolome. As a consequence, different analytical techniques detecting different sets of metabolites may provide sample representations that are different from a biochemical point of view, but that may be equally effective in distinguishing classes of subjects and, then, may help in solving clinical problems.

This was the case of the present study. We proved that the dataset obtained by LF NMR showed the same performance in prediction as the more complex datasets generated by untargeted MS-based metabolomics when used to discriminate between neonates developing sepsis and controls. The chemical space mapped by LF NMR was not the same as that explored by MS. Indeed, due to the strong correlation between LF and HF NMR data, it was possible to interpret the relevant ROIs in terms of metabolites discovering that 2,3,4-trihydroxybutyric acid, 3,4-dihydroxybutanoic acid, d-glucose, d-serine, hippuric acid, lactate, L-threonine, glycine, pseudo uridine, ribitol, kynurenic acid, myo-inositol, taurine and phenylalanine were involved in determining the differences at the urinary metabolic level between neonates developing sepsis and controls. Only taurine and phenylalanine were found relevant both in MS and in NMR analysis. However, in our previous published study [15], glycine and kynurenic acid that were not found to be relevant in urines resulted to be relevant in distinguishing between controls and the EOS group when analyzing plasma samples.

Several studies have been published comparing the urinary metabolome of neonates developing sepsis, EOS or LOS, and controls [24,25,26,27]. However, none were suitable for a clinical application. Interestingly, all the annotated metabolites were discovered also in those studies. Specifically, Fanos et al. [24] discovered 2,3,4-trihydroxybutyric acid, 3,4-dihydroxybutanoic, pseudo uridine and ribitol using GC-MS, and glycine, lactate and d-glucose by HF NMR as related to the differences between healthy neonates and neonates developing sepsis. Serafidis et al. [25] found differences in the level of d-glucose and myo-inositol by HF NMR and in the levels of hippuric acid, phenylalanine and taurine using LC-MS between sepsis and control groups. In Dessì et al. [26], d-serine and L-threonine were discovered using GC-MS. The reader can refer to the references for a detailed discussion about the supposed role played by these metabolites in sepsis development.

Another interesting result of our study is the strong correlation between LF and HF NMR spectra. A similar result was found by Leenders et al. [6] that observed and discussed the close relationship between LF and HF NMR data in the metabolomics investigation of type 2 diabetes.

An important aspect that needs to be investigated by a clinical procedure for decision making is interpretability. Indeed, when an algorithm or, more generally, a classifier is used to make a decision, the manner and reason for the decision that was made must be understood. In other words, the mechanisms underlying decision making should be clear and understandable, and not driven by a black box. This can only be achieved if both the sample representation and the classifier/algorithm are interpretable. The fingerprint based on LF NMR used to represent the sample is in principle interpretable since each ROI can be associated to a single or a set of metabolites and, then, fits the requirement for a clinical application.

The main limitation of our study was that the estimate of the performance in prediction was based on cross-validation and on the out-of-bag predictions during stability selection without an independent test set, because a test set was not included in the experimental design. Moreover, the small number of recruited subjects limits the power of the study. However, this proof-of-concept study is necessary to justify the design of new investigations, where a larger number of recruited neonates is considered and one or more test sets are included to better evaluate the real performance in prediction of the classifier based on LF NMR spectra.

Interestingly, more sophisticated classifiers can be considered when an adequate number of training samples is available. Indeed, we used PLS2C as a classifier to avoid overfitting since PLS-based techniques are suitable for a dataset with a reduced number of observations, and overfitting can be controlled by a randomization test, but random forests may be trained with a larger number of samples and, then, non-linearity and more complex substructures within observations can be taken into account.

## 5. Conclusions

Despite the fact that neonatal sepsis is one of the leading causes of neonatal death and morbidity, there are no effective clinical tools for early detection. Indeed, blood culture is still considered the gold standard, even if false-negative results are not uncommon and the analysis requires a relatively long time. Several metabolomics investigations have been performed in the last 10 years to discover metabolites able to predict EOS, but none have led to suitable markers or a set of markers for clinical application. In the present study, a methodology based on LF NMR fingerprinting is presented and discussed, proving that it is promising for clinical application.

## Figures and Tables

**Figure 1 metabolites-13-01029-f001:**
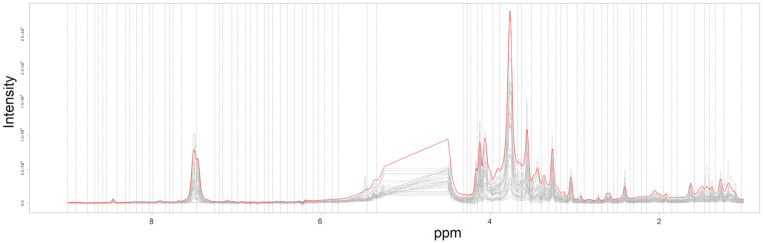
The NMR spectra between 0.05 and 9.00 ppm after alignment (grey lines); the average spectrum used for intelligent bucketing and the intervals used for bucketing the aligned spectra are reported using red line and dashed lines, respectively.

**Figure 2 metabolites-13-01029-f002:**
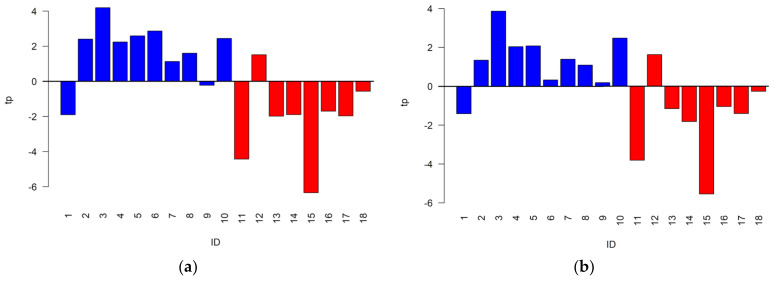
Parallel component tp describing the common data variation discovered by OPLS-W2A between the Fourier 80 dataset and the predictive part of the models obtained considering (**a**) the NEG and (**b**) the POS datasets. Red and blue are used to indicate neonates developing sepsis and controls, respectively.

**Figure 3 metabolites-13-01029-f003:**
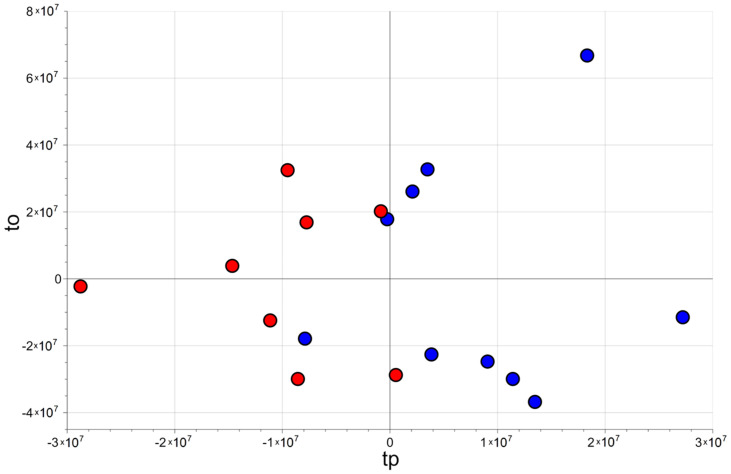
Fourier 80 dataset: score scatter plot of the PLS2C model after post-transformation; tp and to are the predictive and the non-predictive score components, respectively, of the post-transformed model. Controls (blue circles) and neonates developing sepsis (red circles) are separated along the horizontal axis tp.

**Figure 4 metabolites-13-01029-f004:**
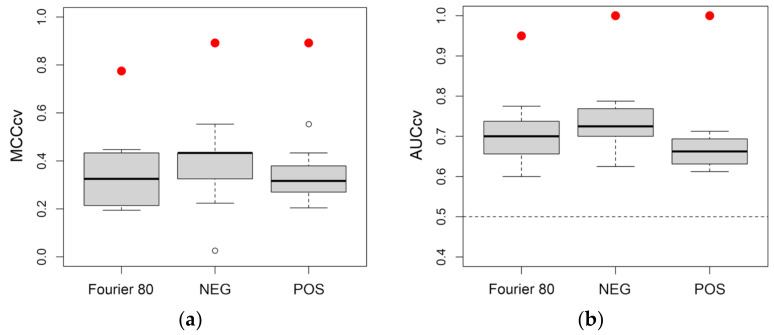
Boxplots summarizing the performance of the PLS2C models obtained with the three datasets: (**a**) distributions of the Matthew’s correlation coefficient calculated by 20 repeated 5-fold cross-validation (MCCcv); (**b**) distributions of the area under the Receiver Operating Characteristic curve calculated by 20 repeated 5-fold cross-validation (AUCcv). Red dots are used to indicate the results in fitting the data.

**Figure 5 metabolites-13-01029-f005:**
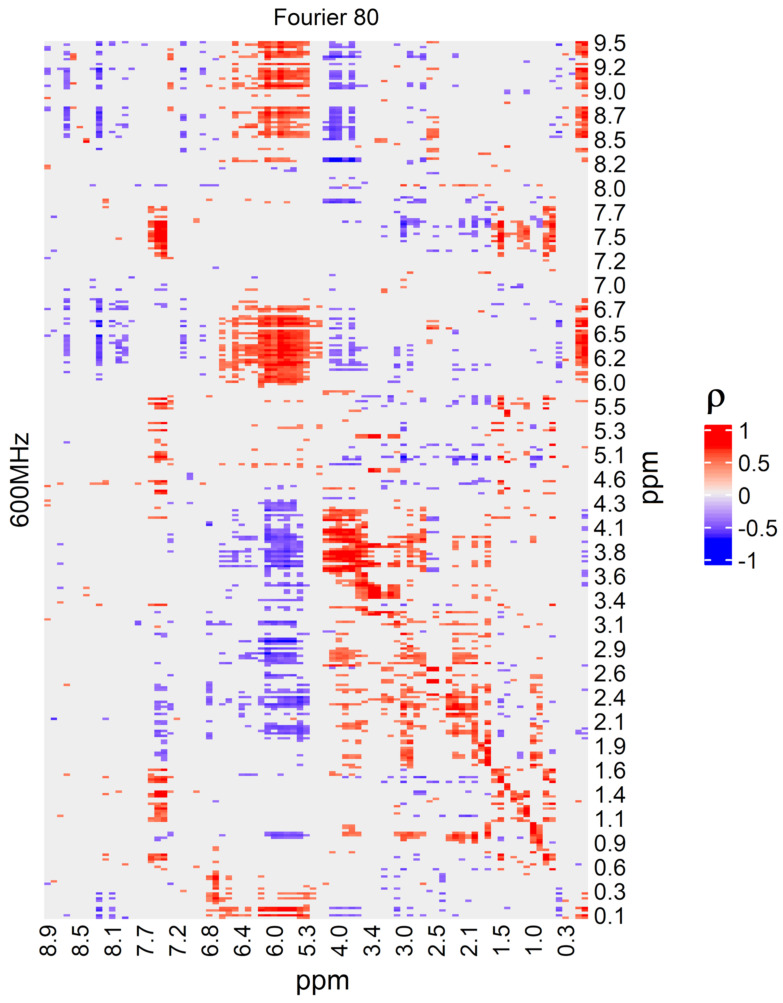
Heatmap based on the Pearson’s correlation coefficient ρ showing the correlation structure between LF NMR (Fourier 80) and HF NMR (600 MHz) spectra. After spectra alignment based on CluPA algorithm, HF NMR spectra were divided into bins by intelligent bucketing with a minimum width of 0.01 ppm and were normalized by PQN. False discovery rate was controlled considering q-values of less than 0.10. Labels indicate the ppm scale.

**Table 1 metabolites-13-01029-t001:** Characteristics of the neonates with urine samples available for the present study; numerical data normally distributed are reported as mean (standard deviation), whereas non-normally distributed data are reported as median [interquartile range], and categorical data are reported as the number of cases (percentage) with respect to the reference group; p is the p-value of the test used to compare the two groups.

Descriptive Variable	Sepsis (*n* = 8)	Controls (*n* = 10)	*p*
Gestational age (days)	210 (16)	213 (14)	0.68
Birth weight (g)	1255 (339)	1396 (360)	0.41
Male sex	2 (25)	3 (30)	0.81
Apgar score 1 min	7 [1.3]	7.5 [0.8]	0.32
Apgar score 5 min	8.2 [1.0]	8.3 [0.8]	0.80
Caesarian section	8 (100)	10 (100)	1.00
Prenatal steroid	8 (100)	10 (100)	1.00
Small for gestational age	2 (25)	2 (20)	1.00
Positive maternal vaginal swab	1 (12.5)	0 (0)	0.44
Premature rupture of membranes > 18 h	2 (25)	2 (20)	1.00
Inotropes	0 (0)	0 (0)	1.00
C-reactive protein—Day 0 (mg/L)	<2.9	<2.9	1.00
White blood count—Day 0 (K/μL)	5.6 [5.3]	9.7 [4.4]	0.08
Platelet count—Day 0 (K/μL)	194 (67)	227 (59)	0.27

**Table 2 metabolites-13-01029-t002:** Relevant ROIs discovered by stability selection: ROI indicates the spectral range of the Region Of Interest in ppm, AUC is the area under the Receiver Operating Characteristic curve, p is the p-value of the Mann–Whitney test and FC[SEPSIS/CTRL] is the ratio between the median of the variable in the group of neonates developing sepsis and that of the controls.

ROI	AUC	*p*	FC[SEPSIS/CTRL]
[7.6541–7.4753]	0.80	0.043	1.52
[7.4753–7.2578]	0.73	0.122	1.36
[5.7942–5.4569]	0.73	0.122	1.46
[5.4569–5.3451]	0.64	0.360	1.40
[4.1537–4.0953]	0.60	0.573	1.51
[4.0953–3.9675]	0.54	0.829	1.11
[3.8876–3.6738]	0.51	0.965	0.97
[3.6277–3.5141]	0.60	0.515	0.80
[3.5141–3.3961]	0.65	0.360	0.88
[3.3961–3.3138]	0.64	0.360	0.88
[3.3138–3.2228]	0.61	0.460	0.89
[1.5835–1.4662]	0.81	0.034	1.46
[1.4662–1.4035]	0.83	0.021	1.49
[1.4035–1.3372]	0.79	0.055	1.44

## Data Availability

The data presented in this study are available on request from the corresponding author as specified in the previously published paper. Data are provided on request because the author must explain how to read the data.

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
