# Peer review of "Low-Field Benchtop NMR to Discover Early-Onset Sepsis: A Proof of Concept"

_metabolites, 2023, doi:10.3390/metabo13091029_

Round 1
Reviewer 1 Report
Dear authors,
First of all, congratulations fro this work. It is really well designed, explained and contextualized...A fantastic group work, that I have no doubts will be high important to metabolomic field.
Secondly, I have no concerns about the metaboilomic study or the statistic approach used in this work. However, at least for me, was not possible evaluate the figure present in the supplementary material (please take a look in the attach bellow, for some reason the figure S1 it is not attached)...maybe was not correctly updated, please check this figure in the final submission, in this way, the further readers will not experience the same situation.
Furthermore, congratulation again for this great scientific work,
Thanks loads,
Cheers,

Author Response
We thank the reviewer for the positive comments.
We have attached a pdf version of the Supplementary Materials where Figure S1 is reported.

Reviewer 2 Report
Low field NMR spectroscopy is a promising approach for express diagnostic of various diseases. From this point of view, the research reported in this manuscript is extremely useful. The main point that the authors themselves note is the restricted number of tests. Nevertheless, the methodology itself is quite interesting, and the results are worthy of publication.
I think that the manuscript can be published after some editing, it is necessary to improve the quality of the drawings (the labels are poorly read, the axes are not always labeled..)
Author Response
We thank the reviewer for the positive comments.
We have checked the english form and edited some figures to improve their quality.
We hope that the current version is suitable for publication.
Reviewer 3 Report
The application of Low-Field benchtop NMR as a diagnostic clinical tool in the case of early-onset sepsis is presented by Matteo Stocchero et al. The authors wrote well the introduction section, updated the data concerning the use of benchtop NMR, presented with a well-defined way the results and finally they discussed them sufficiently.
Minor comment:
A comment on the use of the Low-Field benchtop NMR in lipidomic studies. Are available data in the literature?
Author Response
We thank the reviewer for the positive comments.
There are not many studies about lipidomics based on LF NMR, even if LF NMR may be in principle applied to lipidomics, to characterise particular classes of lipids, for example. An interesting study is Gouilleux et al Anal. Chem. 2019, 91, 3035-3042.
The field of lipidomics is very interesting and our working group is moving towards lipidomics, but the manuscript is dedicated to the investigation of EOS by metabolomics.